# Contamination Levels in Recollected PET Bottles from Non-Food Applications and their Impact on the Safety of Recycled PET for Food Contact

**DOI:** 10.3390/molecules25214998

**Published:** 2020-10-28

**Authors:** Roland Franz, Frank Welle

**Affiliations:** Fraunhofer Institute for Process Engineering and Packaging IVV, Giggenhauser Straße 35, 85354 Freising, Germany; roland.franz@ivv.fraunhofer.de

**Keywords:** post-consumer polyethylene terephthalate (PET), PET bottles, non-food PET applications, PET contaminants, non-intentionally added substances (NIAS), recycling, migration, food packaging, safety, exposure

## Abstract

PET beverage bottles have been recycled and safely reprocessed into new food contact packaging applications for over two decades. During recollection of post-consumer PET beverage bottles, PET containers from non-food products are inevitably co-collected and thereby enter the PET recycling feed stream. To explore the impact of this mixing on the safety-in-use of recycled PET (rPET) bottles, we determined the concentrations of post-consumer substances in PET containers used for a range of non-food product applications taken from the market. Based on the chemical nature and amounts of these post-consumer substances, we evaluated their potential carry-over into beverages filled in rPET bottles starting from different fractions of non-food PET in the recollection systems and taking worst-case cleaning efficiencies of super-clean recycling processes into account. On the basis of the Threshold of Toxicological Concern (TTC) concept and Cramer classification tools, we present a risk assessment for potential exposure of the consumer to the identified contaminants as well as unidentified, potentially genotoxic substances in beverages. As a result, a fraction of 5% non-food PET in the recycling feed stream, which is very likely to occur in the usual recollection systems, does not pose any risk to the consumer. Our data show that fractions of up to 20%, which may sporadically be contained in certain, local recollection systems, would also not raise a safety concern.

## 1. Introduction

Poly(ethylene terephthalate) (PET) bottles are widely used for beverages like soft drinks, mineral water, fruit juices, beers and others. The amount of PET beverage bottles used worldwide continues to increase year by year, a development that is paralleled by an increasing amount of recollected post-consumer PET bottles for recycling. In 2017, for example, on average 58.2% of PET beverage bottles on the market in European countries were recollected and recycled [1]. Indeed, PET beverage bottles have been recycled and safely reprocessed into new packaging applications for more than two decades [2]. The first commercial recycling plant for the production of so-called “super-clean” PET was established in Europe in 1997. Since then numerous recycling processes employing different technologies have entered the market, producing super-clean and food-grade PET recyclates from used (post-consumer) food contact PET materials. In 2008, the European Commission published the Recycling Regulation No. 282/2008, which regulates the use of post-consumer recyclates for direct food contact applications [3]. According to this regulation, recycling companies producing super-clean PET recyclates for such applications require a safety evaluation of their processes by the European Food Safety Authority (EFSA). EFSA has published around 130 opinion letters on the safety evaluation of recycling processes, to date.

In 2011, EFSA published a scientific opinion on the criteria for the safety evaluation of a mechanical recycling process to produce post-consumer recycled PET (rPET) for food contact use [4,5]. The main criterion in the EFSA evaluation approach is the cleaning efficiency of the PET recycling process. Specifically, it considers the technical capability of a process to remove potential contaminants in recollected PET, arising from consumer misuse, down to concentrations that do not pose a risk to human health under the assumption of a worst-case scenario. For this purpose, EFSA applies a highly conservative migration scenario: Starting from a reference contamination level of 3 mg/kg for a contaminant in post-consumer PET, the acceptable maximum migration value for any remaining contaminant in the super-cleaned rPET was set to 0.1 µg/L. The reasoning for these values given by EFSA is that (i) without knowledge of the chemical identity of potential contaminants, any contaminant must be considered a chemical with a structural alert for potential genotoxicity and (ii) infants are consumers of rPET bottled water and are therefore potentially exposed to genotoxic contaminants. To match this migration criterion, the cleaning efficiencies of PET recycling processes must be in the range of at least 90% to 97% for any contaminant, depending on its molecular weight [4,5]. From physicochemical considerations, it is well understood that the removal efficiency of organic-chemical contaminants from a polymer under given time-temperature-pressure conditions depends strongly on the molecular weight of the contaminant. Low molecular weight compounds, like solvents, for example, show significantly higher diffusion rates in PET compared to high molecular weight compounds, like polymer additives [6,7,8,9]. On the other hand, however, the higher diffusion rates exhibited by low molecular weight contaminants lead to their higher sorption into the PET material. In fact, the concentrations of solvents found in post-consumer PET recyclates are significantly greater than those of higher molecular weight compounds [10,11,12]. In other words, the intrinsic chemical properties of a contaminant trigger both the sorption, and therefore contamination level of post-consumer PET, as well as the removal efficiency in the same direction, meaning that higher sorption will be neutralized by higher cleaning efficiency and, conversely, lower sorption by lower cleaning efficiency.

The majority of PET bottles recollected for super-clean recycling have been filled in their first use with water or flavoured beverages. As such, food constituents like limonene, an almost universally used flavour substance, have been determined in nearly all post-consumer PET recyclates [13]. However, PET containers are also widely used in the non-food sector for liquid products, such as shampoos, liquid soaps, mouth washes, sanitary detergents, household cleaners, dishwashing agents, antifreeze solutions, cosmetic skincare milks and others. Therefore, another important aspect in the EFSA opinion [4,5] addresses the source and first use of rPET. Specifically, it states that a percentage of containers used for non-food applications are likely to be present in collection systems of post-consumer PET. This scenario has an influence on the degree of contamination of the input materials for the recycling processes. Notably, contamination can originate from the presence of PET of inferior quality that is not necessarily compliant with the current EU regulation on plastics in contact with foodstuffs, or the sorption of the chemicals from non-food products.

As a consequence, the absorption of product constituents into PET lead to a risk that such non-food PET containers might be contaminated with compounds untypical to food, such as solvents or other ingredients from the product formulations, including potentially hazardous compounds.

The EFSA opinion considers a proportion of 5% or below of PET from non-food consumer applications to be appropriate in the input for recycling [4,5], and deprecates the intentional use of containers from non-food uses as input material. In the case that higher percentages of non-food PET containers are present, the EFSA opinion stipulates that adequate information on the composition of the input is necessary to derive ad hoc figures for a case-by-case evaluation approach, and the petitioner must provide further information to prove the safety of the process. This raises the question and concern as to whether these compounds originating from non-food fillings might be a problem for bottle-to-bottle recycling when PET containers from food and non-food applications are mixed and recycled together. Depending on the proportion in the PET recycling feedstock of post-consumer recyclates sourced from the non-food sector, substances from non-food products might pose a risk for the consumer due to the introduction of such chemicals into the food packaging chain and consequently their potential exposure to the consumer. A proper safety evaluation therefore requires that the proportion of non-food containers entering the recycling stream is known, or conservatively estimated, as are the concentrations of potential non-food contaminants in the PET recyclates. Besides the potential exposure of the consumer towards ingredients from non-food products, a further concern was raised [4,5] as to whether PET materials for non-food applications are of the same food-grade quality as beverage bottles, or whether they contain other, non-authorised constituents, such as additives not listed in EU Regulation No. 10/2011 [14].

Due to the lack of quantitative data on the contamination levels in PET bottles used for non-food applications, EFSA have imposed restrictions on the maximum amount of PET containers from non-food applications in the input of the recycling processes. On the one hand, EFSA recognised that all grades of PET packaging resins sold by European manufacturers and placed on the EU market are food contact grades, even if they are used for non-food packaging applications. Therefore, all additives and processing aids used for the production of PET bottles and containers can be considered to comply with EU Regulation No. 10/2011. On the other hand, however, due to the possible higher absorption of chemical compounds from non-food products bottles, EFSA has limited the amount of PET containers from non-food applications in the input stream of super-clean recycling processes to 5%. Therefore, according to the EFSA guidelines, recycling companies must ensure that the 5% limit for non-food containers is not exceeded [4,5]. Proving this, however, is very difficult for the recycling companies. If, for instance, the input material is baled bottles, the PET containers used for non-food applications can be manually sorted out of the input stream. When recycling companies use washed PET flakes purchased on the market as input materials for their super-clean process, however, it appears questionable or even impossible to distinguish between PET containers previously used for food or for non-food fillings because the labels are removed and the shapes of the bottles are unknown.

To solve this problem, one must go one step back: the reason for the 5% limit recommended by EFSA is the assumption that chemicals from non-food products may be absorbed into the PET bottle wall at higher amounts than food constituents. The missing link here is the knowledge of the concentrations of post-consumer substances in the washed PET flakes originating from non-food uses. This knowledge gap was the motivation of the present study, which had two major objectives:

Determination of the concentrations of post-consumer substances in PET containers used for a range of non-food product applications sourced from the market. Empty containers were collected just before recollection in a green dot system and the containers were analysed qualitatively and quantitatively for any organic compounds present in the container wall material.

Safety evaluation based on the chemical nature and amounts of these post-consumer substances, their potential carry-over into new rPET bottles for beverages, as well as their toxicological Cramer classification for different non-food rPET use level scenarios.

The intention of this publication is to provide such information together with a safety evaluation of using PET from non-food applications as a source for super-clean recycling.

## 2. Results and Discussion

### 2.1. Identities and Concentrations of Substances Found in the PET Containers

Only low molecular weight compounds (up to approximately 200 g/mol) absorbed into the bottle wall of non-food PET containers were found in the present study. Notably, molecular weights up to 250 g/mol represent the crucial range for contaminants in PET that are relevant for migration and potential exposure and therefore for safety evaluation [15]. This is due to the very low diffusivity of larger molecules because of their relatively high activation energies for diffusion [15,16,17,18]. Consequently, larger molecules rarely or even never penetrate into the PET matrix during the first-use life and therefore do not build up a significant potential in and remigration from the polymer matrix. Therefore, substances from non-food products with molecular weights above 200 g/mol were not found in the PET containers. In addition, migration modelling based on realistic diffusion coefficients [18] confirms that the EFSA safety evaluation criterion which is set at a migration limit of 0.1 µg/L would generally be fulfilled for substances with molecular weights above 200 g/mol for long-term migration test conditions. Even after long storage periods of one year at room temperature, this migration limit cannot be exceeded for such substances, thus they are not relevant for safety evaluation.

The 36 investigated PET containers were grouped into six categories according to prior fillings, namely (i) dishwashing detergents, (ii) antifreeze and dish-rinsing agents, (iii) mouthwash products, (iv) kitchen and bathroom sanitary cleaning products, (v) shampoos, and (vi) shower gels and liquid soaps. An overview of the qualitative and (semi)quantitative data of these 36 PET samples is given in Table 1, Table 2, Table 3, Table 4, Table 5 and Table 6. It should be noted that the identities of the bottles could not be clarified in all cases, hence the presence of several unknowns in the tables. In a way, this situation mirrors real-life scenarios, which will always include a certain number of unknowns. Data presented in Table 1, Table 2, Table 3, Table 4, Table 5 and Table 6 include the concentration values for individual samples, as well as the mean value for the whole group. A calculation of mean values for the whole group is in our opinion meaningful because PET recollection systems will always contain a mixture of all these samples and never one sample type only. Following this logic, calculation of overall mean values from the group means would be the next relevant step to predict realistic contamination levels in a fully mixed recollection system. Such overall mean values would form the right basis for further exposure and safety assessment considerations for real-life scenarios. However, for numerous substances this would lead to “dilutions” of concentrations in the full mixture. To avoid this, our approach is to undertake a safety evaluation based on typical concentrations and related ranges found in the six groups. To include also considerations on the potential toxicological concern of the substances identified, Table 1, Table 2, Table 3, Table 4, Table 5 and Table 6 report the Cramer classification results for the substances, including the alert indicators for genotoxic (gtc) and non-genotoxic (ngtc) carcinogenicity.

The first group (Table 1) includes six bottles used for dishwashing products. Only three substances, namely ethanol, phenoxy methanol and phenoxy ethanol, were found in individual bottles at elevated concentrations of 380 mg/kg, 13 mg/kg and 8.4 mg/kg, respectively. The concentrations of all other compounds found in the individual PET bottles within this category were below or far below 10 mg/kg. With the exception of ethylene glycol, which exhibited a low concentration distribution of between 2 and 5 mg/kg in the individual samples, the average concentration values for the mixture of the whole group ranged below or far below 1 mg/kg. Two unknown compounds were found, with group mean concentrations of 0.33 and 0.42 mg/kg. Furthermore, in a few cases wrong assignments by Toxtree were obtained (see footnotes in Table 1), e.g., benzene, a known human carcinogen, were allocated negative gtc and ngtc assignments.

The second category (Table 2) contains four PET bottles originally filled with antifreeze detergents and dishwashing rinsing agents. As expected, the most abundant compound found in these bottles was ethanol, which was present in all four bottles at high levels and exhibited a maximum concentration of 940 mg/kg. High concentrations were also found for other compounds, such as 2-butanone with a concentration of up to 45 mg/kg (sample 8), and concentrations of 23 and 16 mg/kg for two unknown compounds (unknown 3 and unknown 4). The other compounds found in this category, including two further unknowns, were all below 8 mg/kg in the individual bottles. The group means were correspondingly lower, with levels of between 0.5 and 5 mg/kg when excluding ethanol and 2-butanone.

The third group (Table 3) includes six bottles filled in the first use with mouthwash solutions. The presence of flavour compounds in these bottles is quite expected. The most abundant compounds detected were the two cis-trans isomers of anethole, with the highest concentration for the trans isomer of 28 mg/kg in sample 16. Besides two unknowns (7 and 8) found at low levels of below 1 mg/kg, two gtc alerts were obtained from Toxtree for the substances carvone and eugenol, which can be considered as wrong assignments.

Seven bottles were investigated in the category of sanitary cleaning products (Table 4). These bottles contained a variety of different compounds, including seven unknowns. Two samples contained three substances at higher levels: in sample 17, both anethole isomers were found at 31 mg/kg and 44 mg/kg, respectively, and in sample 23 the solvent 2-butanone was quantified at 120 mg/kg. For most of the substances, however, the concentrations were below or far below 8 mg/kg in the individual bottles and the mean values were correspondingly lower. It is noteworthy that the four aldehydes hexanal to nonanal were incorrectly assigned to gtc by Toxtree, and that the substance 2-ethylacrolein, found at 0.6 mg/kg in sample 22 and with a group average level at 0.14 mg/kg, is reported to be genotoxic in vitro (AMES test).

In category 5 (shampoo bottles, Table 5) with six different bottles, only one sample contained ethanol, with a concentration of about 1100 mg/kg. All other compounds found in this category, including two unknowns, were below 9 mg/kg and exhibited mean values in most cases of below 2 mg/kg.

In the last category (Table 6), seven bottles originally filled with shower gels or liquid soap were analysed. Again, ethanol was found in three of the seven bottles at high concentrations of up to 400 mg/kg. The other compounds, including six unknowns, were determined at concentrations below or far below 13 mg/kg in the individual samples, with correspondingly much lower mean values. Benzaldehyde, found at 0.9 mg/kg in sample 33, received an incorrect gtc assignment by Toxtree.

### 2.2. Comparison with Published Data on Contaminant Levels in Post-Consumer PET

Quantitative data on contaminants in post-consumer PET recyclates are rare in the scientific literature. Huber and Franz investigated 22 post-consumer PET flake samples [20] and found that the flavour compound limonene, a ubiquitous substance present in many products, was detected in 15 samples at concentrations of up to 3 mg/kg in the polymer. In one sample, phthalate esters were determined, albeit without full identification and quantification. In a subsequent more comprehensive study, Franz and Welle [21] investigated about 150 post-consumer PET flake samples from food bottles originating from 14 European PET recyclers. The aim was to determine “fingerprints” from headspace gas chromatography/flame ionization detection (GC/FID) analyses to screen for quality and potential migrants. In addition, acetaldehyde and limonene were quantified. The results showed concentrations of between 1.5 mg/kg and 11 mg/kg for limonene and 15 to 40 mg/kg for acetaldehyde, which was quite different from virgin PET (around 1 mg/kg). Since a fraction of the samples were also retrieved from green dot collections, these samples most likely did contain PET containers from non-food applications. Other substances, including those potentially related to non-food fillings, were not targeted in their study. Nevertheless, the GC/FID fingerprints revealed the presence of such substances, albeit at peak intensities that were comparable to or lower than limonene. This indicated that other substances were present at the same concentration range as limonene, i.e., up to approximately 10 mg/kg.

In the USA, Bayer investigated PET materials (flakes superficially cleaned by a commercial wash process) from five different post-consumer PET feed streams for bottle-to-bottle recycling processes [22] taken from deposit and curbside collections. Whereas the investigated deposit collections contained nearly 100% food containers, the amount of non-food containers in curbside collections ranged from 0.04% to 6%. From one of these curbside collections a fraction of non-food use containers was sorted out for this study. Bayer identified and semi-quantified 121 compounds absorbed in PET. The total concentration of these compounds was approximately 28 mg/kg in deposit bottles (nearly 100% PET bottles previously used for food applications) and 39 mg/kg in PET bottles used for non-food fillings, respectively. The key substances found in both fractions were, from a concentration standpoint, the flavour compounds hexanal, benzaldehyde, limonene, methyl salicylate, and carvacrol. These data for PET containers from non-food applications are in good agreement with the concentrations found in the present study with the exception of ethanol and 2-butanone, which were not found by Bayer. The most likely reason for this absence is that Bayer analysed the post-consumer PET samples after a commercial washing process, which included a hot washing step for about 10 min at temperatures of about 85 °C with caustic soda and detergents followed by a drying step, which significantly reduces the concentrations of volatile solvents like ethanol. Published studies involving surrogate contamination have shown that simple washing and drying reduce the concentrations of volatile contaminants in PET by at least 98%, and even non-volatile contaminants by 79% [23,24].

Begley et al. [10] investigated the sorption kinetics of hexachloro cyclohexane (lindane) from a commercial medical shampoo into PET materials, one being amorphous sheet material and the other material from PET bottle wall at 20 °C and 40 °C. The amount absorbed into PET from the shampoo after 231 days was 28 mg/kg in the sheet material at 20 °C and at the same level in the bottle material at 40 °C. However, the concentration in the sheet material at 40 °C reached a surprisingly high level of 745 mg/kg. This can be explained by the fact that the shampoo contained 7% acetone, a very aggressive solvent that causes swelling of the amorphous PET sheet material. As a consequence, swelling of the PET matrix triggers higher sorption of the shampoo ingredient (the formulation contained 1% lindane) under the exaggerated sorption conditions at 40 °C. In our opinion, this high value cannot be considered as a potentially realistic contamination level. Firstly, PET containers and amorphous sheet material are not equivalent, and the latter are typically used for non-food products. Secondly, such a long-term contact at 40 °C is not a realistic storage scenario. Furthermore, from analysis of selected non-food PET bottles obtained directly from collection bins for recycling plastic bottles in the USA, the authors [10] found 130 mg/kg to 204 mg/kg of methyl salicylate in a antiseptic mouthwash bottle, 1 mg/kg of the antibacterial agent 5-chloro-2-(2,4-dichlorophenoxy)-phenol (trade name Triclosan) in a PET bottle used for hand soap, as well as 1.6 mg/kg of the flavour compound limonene in a paint remover bottle. The values for Triclosan and limonene are in very good agreement with our findings; however, we found methyl salicylate only in one mouthwash bottle and at a level of more than 100 times lower (Table 3). The reason for this may be due to a strong interaction of ethanol with PET, which enhances the uptake of methyl salicylate into the polymer. Unfortunately, the concentration of ethanol was not determined in the aforementioned study, although a level of more than 1000 mg/kg would be expected in view of the reported parameters. In our case, ethanol was not detected (sample 13). This is likely due to the fact that modern packaging design aims to achieve better compatibility between fillings and packaging materials, with the effect that sorption of product ingredients will be minimized.

Franz et al. [13] published data from a Europe-wide screening of washed post-consumer PET flakes. Overall, 689 different post-consumer PET flake samples were collected from twelve European countries and were analysed regarding for post-consumer contaminants. Since one sample consisted of 10–15 flakes, each originating from a different bottle, it was concluded that some 7000–10,000 bottles were included in this screening project. In contrast to Bayer [22], the authors did not have the information needed to distinguish between post-consumer PET bottles from food contact applications and non-food containers, thus could not determine the fraction of non-food containers, if any. However, it is most likely that also small amounts of PET containers from non-food applications were present among the investigated samples, depending on the recollection system. The proportion of such PET containers, however, was not available. As a result, mean concentrations for the most prominent substance limonene (flavour in soft drinks) and acetaldehyde (PET typical degradation substance) in rPET flakes were reported to be 2.9 mg/kg and 18.6 mg/kg, respectively, with maximum concentrations of 20 mg/kg and 86 mg/kg, respectively. In addition, several substances untypical of PET, such as phthalates, adipates and erucamide, were found at levels from 0.05 mg/kg to 0.2 mg/kg, with a maximum concentration of 0.5 mg/kg for the substance dioctyl phthalate. Due to the sample characteristics mentioned above (10-15 flakes per sample measured), this translates to approximately a tenfold higher concentration in the particular flake and therefore in the associated bottle, i.e., to a concentration range from 0.5 mg/kg up to 5 mg/kg. It cannot be established whether or not these substances originated from non-food fillings or from carry-over from other sources, such as labels, sleeves, cups, coatings, etc., but it provides an indication that these values are in congruence with the measured values of this study. In any case, due to the fact that the investigated post-consumer PET flakes were drawn from the market, they most probably also contained some material from non-food PET containers.

Dutra et al. investigates also recycled FET samples. Several post-consumer substances were identified in post-consumer rPET flakes and pellets samples. The list of identified substances is in agreement with our findings. However, no quantitative data are given in this study [25].

As a conclusion here it can be stated that both our data and the reported literature data, indicate that levels of contamination from non-food applications can be found from the sub-mg/kg range up to around 30 mg/kg when we exclude solvents of a very volatile character, such as ethanol and 2-butanone. Concerning our data presented here, by far the highest fraction is found in the range below 10 mg/kg for individual bottles, and hence below 1 mg/kg for the group mean values.

### 2.3. Estimation of the Fraction of Non-Food Products PET in the Recycling Feed Stream

The estimation of the fraction of non-food products PET in the recycling feed stream is a very difficult undertaking. Many producers of super-clean recycled PET purchase washed PET flakes produced from ground PET bottles by recycling companies that only run washing lines but not super-clean recycling processes. Use of such PET flakes as input materials make it nearly impossible to determine the amount of the PET fraction previously used for non-food applications, since PET flakes from food and non-food applications are indistinguishable. Considering the step prior to grounding into flakes, i.e., the recollected bottles compressed into compact bales, in principle it is possible to distinguish between the two, but due to the high compression of the baled bottles the fraction of non-food bottle might be nevertheless difficult to be determined. However, since quantitative data about the amount of non-food containers in the input stream and their contamination levels are required for the evaluation of compliance with EFSA criteria, recyclers need to find a way to perform an analytical sorting. Such proprietary data are not usually available for the different recollections systems and could only be estimated by the particular recycler based on opening the bales and counting the different bottles. However, such a laborious and time-consuming procedure will result only in a snapshot of the actual input material and not substantiate a general picture. Furthermore, the fraction of non-food PET will not be consistent in different feed streams. Due to the different recollection systems in the European member states, it is quite obvious that the fraction non-food PET containers might be very different, whereby the percentage may range from virtually zero for deposit bottles systems up to a higher percentage for curbside systems and more general plastics recollections. Supporting data are very rare in the scientific literature. Begley et al. [10] mention that 20% of the PET containers foreseen for PET recycling in the USA might be non-food product bottles. However, this seems to be a rough estimate from the recollection figures from 1999 and not necessarily current for the situation today.

In the following, an attempt is made to approximately estimate the maximum amount of PET containers from non-food applications based on the available market data. The PET resin market can be divided into two main application fields: (i) the fibre applications and (ii) the packaging applications, e.g., PET containers and sheet material. In 2008, the worldwide market size was 45 million tonnes of PET resins, from which 63% were for fibre and yarn applications that are not collected in packaging recollection systems and therefore can be excluded from these considerations. From the non-fibre PET resins, 35% were used for bottling of carbonated soft drinks, 25% for mineral water, and 17% for other beverages, which amounts to 77% used overall for the beverage bottles market. Furthermore, 9% of the PET resins were used for other food packaging applications, 10% for films and sheet, and 4% for other (non-food) packaging application, which amounts to 23% not used for the beverage bottle market. We consider (i) that the input materials for PET bottle-to-bottle recycling processes are typically PET containers and (ii) that sheet/film materials are typically not collected or separated during recycling (in the first washing step) so that 10% sheet material does not enter the recycling process. We also exclude (iii) the 9% fraction for other food from recycling and assume (iv) that the full 4% fraction for other packaging goes into the category non-food products packed in PET containers. We assume conservatively that this 4% fraction enters the recycling feed stream together with the 77% beverage bottle material, and that both fractions together make up 100% of the recycling feedstock. This recycling feed stream scenario would then contain 5% non-food packaging and 95% beverage bottles. This, of course, would be an average scenario of all recollection systems and could vary, as already stated and within certain limits, from one recollection system to the next. In conclusion, scenarios from very low percentages (<1%), as most likely originating from deposit systems, up to percentages of 5% or even somewhat higher, as possibly originating from less controllable curbside recollections, are estimated. However, a percentage of 20%, as mentioned above, appears unlikely from the reported market data but may occur sporadically under specific local recollection conditions.

### 2.4. Evaluating the Impact of the Contamination Levels in Non-Food PET Applications on the Safety of Recycled PET Bottles for Food Contact

Regarding the impact of the contamination levels in non-food containers on the safety of rPET in direct food contact, the question of interest is: Which concentrations of which substances from non-food PET containers can be established in beverages filled in PET bottles made from rPET based on usual super-clean recycling processes? Or, in other words: Which amounts of such non-food substances would remain in the rPET bottle walls after the reprocessing conditions and migrate from there into the beverages, typically after one year at room temperature (25 °C), as defined by EFSA as typical storage conditions for mineral water bottles? Further: How toxicologically relevant is the potential exposure of the consumer to these concentrations in beverages? An answer to these questions can be most likely only given by designing a conservative scenario of the situation in combination with mathematical modelling of the underlying diffusion and migration processes and assisted by the so-called Threshold of Toxicological Concern (TTC) concept [26]. We therefore assume the following scenario: A 5% fraction of non-food PET is processed together with 95% post-consumer PET bottle material into new beverage bottles in a typical super-clean recycling process. The contaminants in the non-food PET after surficial washing and rinsing, which correspond to usual commercially washed rPET flakes used as input for super-clean processes, are represented by the substances listed in Table 1, Table 2, Table 3, Table 4, Table 5 and Table 6. The cleaning efficiency of the super-clean process is assumed to be 90% for any contaminant, which is the typically lowest efficiency of such processes, usually ranging between 99% and 90%, depending on the substance; cleaning efficiencies of only 90% can be considered as worst-case for PET super-clean recycling processes. The rPET produced in this manner is then manufactured into beverage bottles (for simplicity reasons) with a surface-to-volume ratio of 6 dm^2^ per litre beverage. Migration into the beverage is modelled after contact conditions of one year at 25 °C (room temperature). Two generic migration curves derived from these assumptions are presented in Figure 1. The figure depicts the migration into the beverage as a function of the molecular weight of contaminants using two different prediction models for the diffusion coefficients of potential migrants. These generic curves are based on the assumption of a starting contamination level of 10 mg/kg in the non-food PET, which is reduced by a factor of 20 due to the assumption of only a 5% fraction of non-food PET. This results in a concentration of 0.5 mg/kg. The concentration is further decreased by 90% due to the cleaning efficiency of the super-clean process, leading to a contamination level of 0.05 mg/kg in the final rPET bottle. The migration is then modelled using this 0.05 mg/kg as the initial concentration in the polymer, the so-called c_P,0_ value, for the migration model for the above-mentioned contact conditions of one year at 25 °C.

Of the two prediction models depicted in Figure 1, the rather conservative Piringer A_P_ model, which is described in [27,28,29], is generally accepted and applied in many EFSA evaluations of recycling processes. The other model uses the Welle equation [9], which gives a more realistic estimate of the diffusion in the PET polymer. From these two generic curves, using 5% non-food PET and 10 mg/kg input contamination level in non-food PET as defined generic starting points, the concentrations of other substances in the beverage can be easily derived for any other non-food PET contamination level and/or percentage in the feed stream because of the linear relationship of the migration value of c_P,0_ and the percentage used. When we increase the fraction of non-food PET up to 20%, these values increase each by a factor of four compared to a non-food container level of 5%. On the other hand, when we decrease the concentration in the non-food PET from 10 mg/kg to 0.5 mg/kg, the migration decreases by a factor of 20. A combination of both will lead to a reduction by a factor of five.

As an example: for the substance carvone (molecular weight 150 g/mol), if present at 10 mg/kg in the non-food feed stream, migrations of 0.04 µg/kg and 0.001 µg/kg can be derived from the generic curves of the Piringer and the Welle model, respectively. When considering the value measured in sample 16, which is 2.4 mg/kg (see Table 3), the related migration value would be roughly a factor 4.17 lower, i.e., 0.009 µg/kg and 0.00024 µg/kg, respectively. Based on the mean group value of samples 11–16, however, the migration would be a factor 25 lower, i.e., 0.0016 µg/kg and 0.00004 µg/kg, respectively. The curves in Figure 1 allow a very quick and straightforward safety evaluation for all substances identified with no genotoxic alert, as listed in Table 1, Table 2, Table 3, Table 4, Table 5 and Table 6. With the exception of ethanol, migration will in all cases be lower than 1 µg/kg and, in most cases, even far (one to several orders of magnitude) below 1 µg/kg. Therefore, regardless of the Cramer class assignment of a compound, either I, II or III, it can be considered to be safe. As mentioned above, ethanol and other small and very volatile molecules such as 2-butanone are determined in higher concentrations as 10 mg/kg. These very volatile substances will be removed by the super-clean process at cleaning efficiencies equal to or higher than 99% and are therefore not further considered here.

A special group of contaminants in non-food containers are substances with genotoxic and carcinogenic potential, and these require a separate discussion, as follows: In Table 1, Table 2, Table 3, Table 4, Table 5 and Table 6, several substances are listed for which the Toxtree software assigns a genotoxic or non-genotoxic carcinogenicity (gtc or ngtc) alert. Examples include aliphatic aldehydes, benzaldehyde, eugenol and others. These assignments are evidently incorrect and are not further discussed here. Two substances remain, namely benzene, listed in Table 1, which is a known human carcinogen (but not gtc or ngtc assigned by Toxtree), and 2-ethylacrolein, which is known to be mutagenic in vitro [19]. Both substances were found at group means levels of 0.15 mg/kg and 0.14 mg/kg, respectively. Benzene and ethylacrolein have similar molecular weights of 78.1 g/mol and 84.1 g/mol, respectively. In terms of migration, these values would correspond to concentrations in a beverage of 0.00107 µg/kg (benzene) and 0.00101 µg/kg (2-ethylacrolein) for the Piringer prediction model and 0.000338 µg/kg (benzene) and 0.000240 µg/kg (2-ethylacrolein) for the Welle prediction model. These values were calculated for an input fraction of 5% non-food PET and a cleaning efficiency of 90% for the super-clean process.

The more conservative value (Piringer) corresponds to a dietary exposure of benzene of 0.000161 µg per kg body weight (bw) per day for an infant drinking 0.75 L of water, which is the crucial scenario applied by EFSA [4,5]. This is a factor 15.5 lower than the EFSA “benchmark” criterion from TTC concept of 0.0025 µg per kg bw per day applied to any potential genotoxic contaminants migrating from rPET food contact materials (FCMs). From the more realistic model (Welle), a dietary exposure of benzene of 0.0000507 µg per kg bw per day follows, which is a factor 49.3 lower than the EFSA benchmark value.

It is interesting to note that a recent publication [30] reported migration values for benzene from rPET bottles into mineral water of 0.03 µg/L to 0.44 µg/L. The level of concern for public health was evaluated based on the margin of exposure (MOE) concept [31,32]. The reported MOE values ranged from 3 × 10^5^–8 × 10^6^ and, being considerably higher than 10^4^, were considered to be of low concern. In our case, the modelled migration values are at least one order of magnitude lower, which increases the MOE correspondingly. In conclusion, the expectable level of concern for benzene and 2-ethylacrolein can be considered at most as extremely low.

The final question to be discussed and answered is how to deal with the unknowns, which in principle could consist partly or even totally of genotoxic substances? The “sanitary cleaning products” group (see Table 4) exhibited the highest number of unidentified substances (unknowns 9–15), amounting to seven compounds with group mean levels below of 1 mg/kg and with estimated molecular weights of between 130 and 160 g/mol. When assuming that all of these seven substances are genotoxic, the following simple consideration can be made: Summing up the concentrations of the seven substances gives 1.82 mg/kg, which we round up to 2 mg/kg. We simplify the approach by reducing these seven substances to only one, with a mean molecular weight of 145 g/mol and a concentration of 2 mg/kg. A migration (Piringer) of 0.0418 µg/kg can be derived from the generic migration curve in Figure 1 (Piringer), which yields 0.00835 µg/kg (when divided by 5; c_P,0_ is only 2 mg/kg). From this, the exposure for an infant drinking 0.75 L of water would be 0.00125 µg per kg bw per day, which is a factor 2 below the EFSA “benchmark” criterion; by comparison, the Welle prediction model yields an exposure of 0.0000360 µg per kg bw per day, which is a factor 69.4 lower. From these results, even assuming that all unknowns are genotoxic (which can be reasonably assumed not to be true), the expectable level of concern, if any, can be considered very low.

As mentioned above, Begley et al. [10] debated a potential fraction of 20% PET non-food product bottles in the recycling feed stream, which according to our approach based on reported market data appears to be unlikely. Nevertheless, when assuming 20% non-food PET, all safety margins derived above for the 5% non-food fraction would decrease by a factor of 4 due to the linear proportionality of the calculated migration (Figure 1) with the initial content of contaminants in the PET material. For instance, the safety factor of 69.4 derived above from a genotoxicity assumption for all seven unknowns would decrease by a factor of 4, resulting in a fourfold higher exposure that is still lower than the EFSA “benchmark” criterion by a factor 17.4. This indicates that even a 20% non-food PET fraction would not pose a significant level of concern for the health of consumers.

## 3. Materials and Methods

### 3.1. Collection of PET Containers

A total of 36 PET containers used for six different non-food product categories present on the German market were collected at the end of use in an empty state from consumers. These containers are usually recollected for recycling together with other post-consumer plastics by the German green dot system. Accordingly, these PET containers can be considered as representative input materials for commercial recycling. Due to the recollection directly from consumers, cross-contamination with other products in the green dot recollection system could be excluded. In fact, the intention of this study was to explore the presence of substances from the non-food products themselves, rather than from post-collection cross-contamination. The empty containers, which occasionally contained very small amounts of residual liquids, were rinsed with a 10% ethanol solution in water to clean the inner surfaces and remove surface contaminants that had not been absorbed into the PET material. It should be noted here that the surface contaminants are not of interest because they are typically readily removed by the washing processes in one of the first cleaning steps of any recycling process.

Within this study, the containers (*n* = 36) were divided into the following six liquid non-food product categories: (i) dishwashing detergents (*n* = 6), (ii) antifreeze and dish-rinsing agents (*n* = 4), (iii) mouthwash products (*n* = 6), (iv) kitchen and bathroom sanitary cleaning products (*n* = 7), (v) shampoos (*n* = 6), and (vi) shower gels and liquid soap products (*n* = 7).

### 3.2. Quantitative and Semi-Quantitative Determination of Chemicals in the PET Containers

A high temperature headspace gas chromatography-based method was applied to quantify the chemicals in the PET containers. The same approach was previously used to screen washed PET recyclates within the European “Recyclability” Project [13]. The method has not been changed to ensure the comparability of results. 1.0 g of the sample was analysed using headspace gas chromatography with flame ionisation detection (FID). Gas chromatograph: Perkin Elmer (Rodgau, Germany) AutoSystem XL; column: DB 1 - 30 m - 0.25 mm inner diameter (i.d.) - 0.25 µm film thickness; temperature programme: 50 °C (4 min), rate 20 °C/min, 320 °C (15 min); pressure: 50 kPa helium; split: 10 mL/min. Headspace autosampler: Perkin Elmer HS 40 XL, oven temperature: 200 °C, needle temperature: 210 °C, transfer line: 210 °C, equilibration time: 1 h, pressure build-up time: 3 min, injection time: 0.02 min, withdrawal time: 1 min.

Identification of the volatile compounds was done by a coupling of headspace GC with mass spectrometry (MS): (Perkin Elmer (Rodgau, Germany) Clarus 600), column: ZB 1 MS guardian - 30 m - 0.25 mm i.d. - 0.25 µm film thickness; temperature programme: 40 °C (2 min), heating rate 10 °C/min, 320 °C (5 min). Headspace autosampler: oven temperature: 200 °C; needle- and transfer line temperature: 210 °C; equilibration time: 1 h; pressure build-up time: 3 min; injection time: 0.04 min; withdrawal time: 1 min. Mass spectrometer: electric ionisation, full scan mode with mass range *m*/*z* 35–300. The applied method can detect compounds in PET up to a molecular weight of about 250 g/mol with detection limits of below 1 mg/kg.

Quantification of possible contaminants in non-food PET bottles was achieved by comparison of the peak detector response of determined substances and of a range of *n*-alkanes introduced into a PET bottle wall during PET preform production. The concentrations of the *n*-alkanes were determined quantitatively by extraction with dichloromethane followed by gas chromatographic detection. The *n*-alkanes were quantified via the standard addition method. 1.0 g of the samples was extracted with 10 mL of dichloromethane for 3 days at 40 °C by total immersion. An internal standard of butylated hydroxyanisole (BHA) und Tinuvin 234 was added to an aliquot of the extracts, and analysed by gas chromatography with flame ionisation detection (Agilent (Wilmington, USA) 6890) for semi-volatile compounds. The extraction solutions were analysed by gas chromatography with flame ionisation detection (GC-FID): column: DB-1 - 20 m, 0.18 mm i.d. - 0.185 µm film thickness; temperature programme: 50 °C (2 min) heating rate of 10 °C/min, 340 °C (10 min).

The volatile compounds ethanol, 2-butanone, ethyl acetate, ethylene glycol, tetrahydrofuran and benzene were quantified by multiple headspace extraction according to the procedure published by Kolb [33,34]. 1.0 g of the sample was analysed with multiple headspace extraction (MHE) with six injections. Gas chromatograph: Perkin Elmer AutoSystem XL, column: DB 1 - 30 m - 0.25 mm inner diameter (i.d.) - 0.25 µm film thickness; temperature programme: 50 °C (4 min), rate 20 °C/min, 320 °C (15 min); pressure: 50 kPa helium; split: 10 mL/min. Headspace autosampler: Perkin Elmer HS 40 XL; oven temperature: 200 °C; needle temperature: 210 °C; transfer line: 210 °C; equilibration time: 1 h; pressure build-up time: 3 min; injection time: 0.02 min; withdrawal time: 1 min.

### 3.3. Migration Modelling

The migration modelling was performed using the AKTS SML software version 4.54 (AKTS AG Siders, Switzerland). The program uses finite element analysis. The associated mathematical procedure and the main equations have been published by Roduit et al. [35].

### 3.4. Cramer Classification and (Non)Genotoxic Carcinogenicity Alert

Cramer classification of the identified substances was performed by Toxtree (Toxic Hazard Estimation by decision tree approach), freely downloadable from Toxtree [36]. Alerts for genotoxic or non-genotoxic carcinogenicity (gtc or ngtc) were determined using the integrated decision tree “Carcinogenicity (genotoxic and non-genotoxic) and mutagenicity rule base from ISS”.

## 4. Conclusions

PET recycling for new food contact has several implications. First, it is not possible to distinguish between beverage bottles and non-food containers in the ground bottle flakes. In baled bottles, however, this should be possible, in principle, although their high compression makes the fraction of non-food bottles difficult to quantify, if possible at all. Second, most recycling companies use purchased washed flakes as input materials for their super-clean recycling processes, with the same consequence that the amount of PET flakes from non-food containers remains largely unknown. Correspondingly, substantial quantitative data on the amount of non-food containers in the PET recycling feed stream, as well as knowledge of the associated contamination levels, are required in order to evaluate whether rPET bottles comply with EFSA safety criteria. From the market data available, we conclude that the PET recycling feed stream for PET bottle-to-bottle recycling may contain up to 5% non-food containers as an average over all recollection systems. A percentage of 20% appears unlikely from the reported market data, but may occur sporadically under specific local recollection conditions. The actual proportion of non-food PET will depend on the type of the particular recollection system. Deposit recollections will contain significantly lower percentages of non-food containers, whereas curb-side collections are likely to contain higher percentages.

In terms of the type, nature and concentrations of non-food product ingredients found in non-food PET containers in this study, the following conclusions can be drawn: based on the results of our study, the reported literature, and considering the inherent diffusivity properties and high chemical inertness of PET, it can be concluded that the levels of contamination in mixed non-food PET collections are generally expected to range from sub-mg/kg concentrations up to around 30 mg/kg when excluding very volatile solvents, such as ethanol, 2-butanone and other. The reason for the latter exclusion is that these compounds are efficiently (99%) removed during usual super-clean processes. Our data presented here show that the highest fraction of contaminants by far is found in the range below 10 mg/kg for individual bottles, with corresponding levels for mixed collections likely to be always below 1 mg/kg.

With the exception of benzene and 2-ethylacrolein, all of the identified substances could be assigned to Cramer classes I, II or III. Based on the assumption of 5% non-food PET in the feed stream and applying migration modelling, the exposure of the consumer to all substances will be far below the corresponding TTC levels, meaning that these substances do not raise any safety concern even for 20% non-food PET starting fraction. Assuming that the unidentified substances also found and semi-quantified in non-food PET are genotoxic, then the estimated exposures are below the EFSA criterion derived from the lowest TTC level for genotoxic carcinogens by factors 2 or 69.4 (depending on the migration model). For the 20% non-food PET scenario, these safety margins are reduced, yet are still a factor of 17.4 lower than the EFSA criterion when using the more realistic migration model.

Our findings and conclusions confirm the safety criterion of accepting 5% fraction of non-food PET applications in the feed stream, as set in the EFSA guidelines. This criterion appears to be very conservative, however, and is not necessary from a migrational point of view, and thus could be disregarded when taking into account that controlling the proportion of non-food PET is often not possible and that even fractions of up to 20% would not increase the levels of concern for consumer health from rPET bottles.

## Figures and Tables

**Figure 1 molecules-25-04998-f001:**
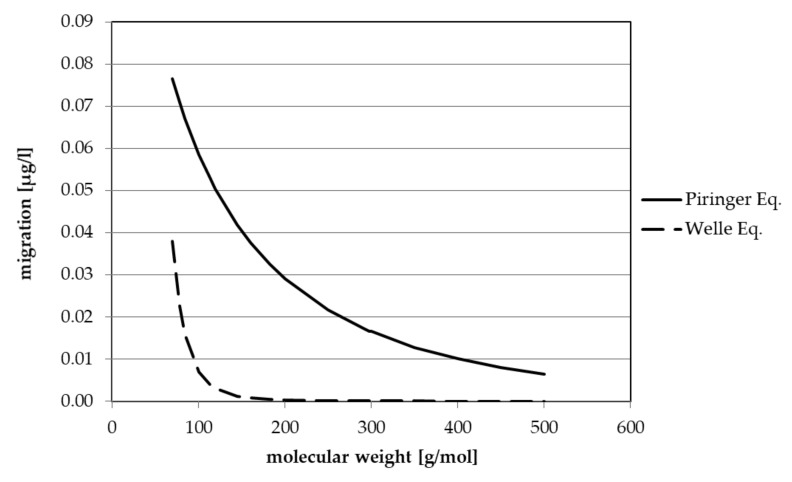
Correlation between the migration and the molecular weight calculated from predicted diffusion coefficients (PIRINGER Eq. [25,26,27] and WELLE Eq. [8] with an concentration of 10 µg/kg in the PET non-food input with 5% non-food content and a cleaning efficiency of 90% after storage for 365 day at 25 °C (EU cube)).

**Table 1 molecules-25-04998-t001:** Substances found in PET bottles originally used for dishwashing products.

Substance	GC R_t_	Cramer Classification					Concentration in PET [mg/kg]						
(mol.weight [g/mol])-	[min]	Class			Alert		Sample No.						
*CAS-No.*		I	II	III	gtc	ngtc	1	2	3	4	5	6	Mean 1–6
Ethanol (78.4)-*64-17-5*	1.8	x			-	-				380			63.3
Unknown 1	1.8	?	?	?	?	?						2.0	0.33
2-Butanone (79.6)-*78-93-3*	2.1	x			-	-				7.5			1.25
2-Methyl-1,3-dioxolane (88.1)-*497-26-7*	2.4	x			-	-	0.4	1.5	1.8	0.3	1.0	0.8	0.97
*n*-Butanol (74.1)-*71-36-3*	2.5	x			-	-		0.5					0.08
Benzene (78.1)-*71-43-2*	2.5			x^*)^	-^*)^	-^*)^					0.9		0.15
Ethylene glycol (62.1)-*107-21-1*	2.7	x			-	-	2.1	5.1	3.0	3.3	3.2	3.1	3.3
Toluene (92.1)-*108-88-3*	3.9			x	-	-				2.4			0.4
*o*-Xylene (106.2)-*95-47-6*	5.6			x	-	-			0.3				0.05
*m*-Xylene (106.2)-*108-38-3*	5.8			x	-	-			0.6				0.1
Benzaldehyde (106.1)-100-52-7	5.8	x			+^*)^	-				0.6			0.1
*p*-Xylene (106.2)-*106-42-3*	6.1	x			-	-			0.1				0.02
Acetic acid hexylester (144.2)-*142-92-7 (n)* or *628-95-5 (iso)*	7.8	x			-	-				0.7			0.12
Limonene (136.2)-*7705-14-8*	8.1	x			-	-	0.7		1.6				0.38
Nonanal (142.2)-*124-19-6*	8.8	x			+^*)^	-	0.4		0.2	0.3		0.3	0.2
Unknown 2	8.9	?	?	?	?	?		2.5					0.42
Benzoic acid (122.1)-*65-85-0*	9.3	x			-	-				3.3			0.55
Acetic acid benzylester (150.2)-*140-11-4*	9.3	x			-	-					2.6		0.43
Decanal (156.2)-*112-31-2*	9.8	x			+^*)^	-	0.2	0.2	0.4	0.2			0.17
Phenoxymethanol (124.1) or	9.8			x	-	-						13	2.2
Salicylalcohol (2-hydroxymethyl phenol) (124.1)-*90-01-7*		x			-	-							
2-Hydroxy-4-methyl benzaldehyde (136.2)-*698-27-1*	10.1	x			+^*)^	-					0.4		0.07
1-Phenoxy ethanol (138.2)-*122-99-6*	10.1		x		-	-						8.4	1.4

x: Cramer classification; ?: classification not possible due to unknown substance; -: negative alert; +: positive alert; ^*)^ wrong assignments by Toxtree: benzene is a known human carcinogen; benzaldehyde is not a human carcinogen and not genotoxic; nonanal and decanal: not human carcinogens, JECFA/DG Sante authorized food flavouring agents; 2-hydroxy-4-methyl benzaldehyde = 4-methyl salicylaldehyde: EU Food Additive—Flavis no. 05.091.

**Table 2 molecules-25-04998-t002:** Substances found in PET bottles originally used for antifreeze and dishwashing products.

Substance	GC R_t_	Cramer Classification					Concentration in PET [mg/kg]				
(mol. weight [g/mol])-	[min]	Class			Alert		Sample No.				
*CAS-No.*		I	II	III	gtc	ngtc	7	8	9	10	Mean 7–10
Unknown 3	1.7	?	?	?	?	?			23		5.75
Unknown 4	1.7	?	?	?	?	?			16		4.0
Ethanol (78.4)-*64-17-5*	1.8	x			-	-	710	940	180	90	480
Unknown 5	2.1	?	?	?	?	?			7.3		1.83
Acetic acid (60.1)-*64-19-7*	2.1	x			-	-				3.2	0.80
2-Butanone (79.6)-78-93-3	2.1	x			-	-	21	45	5.2		17.8
Unknown 6	2.2	?	?	?	?	?			2.2		0.55
2-Methyl-1,3-dioxolane (88.1)-*497-26-7*	2.4	x			-	-	1.0	1.3		0.4	0.68
Ethylene glycol (62.1)-107-21-1	2.7	x			-	-	5.9	12		1.4	4.83
Limonene (136.2)-*7705-14-8*	8.1	x			-	-	3.2	3.7			1.73

x: Cramer classification; ?: classification not possible due to unknown substance; -: negative alert; +: positive alert.

**Table 3 molecules-25-04998-t003:** Substances found in PET bottles originally used for mouthwash products.

Substance	GC R_t_	Cramer Classification					Concentration in PET [mg/kg]						
(mol.weight [g/mol])-	[min]	Class			Alert		Sample No.						
*CAS-No.*		I	II	III	gtc	ngtc	11	12	13	14	15	16	Mean 11–16
Ethanol (78.4)-*64-17-5*	1.8	x			-	-						240	40
Tetrahydrofuran (72.1)-*109-99-9*	2.3		x^*)^		-	-						0.4	0.07
2-Methyl-1,3-dioxolane (88.1)-497-26-7	2.4	x			-	-	0.5	1.9	0.3	1.1	0.7	0.7	0.87
Ethylene glycol (62.1)-*107-21-1*	2.7	x			-	-	3.4	3.2	2.3	6.6	3.9	5.6	4.17
2,2-Dimethyl-1,3-dioxolane (102.1)-*2916-31-6*	5.0			x	-	-		7.8					1.3
Limonene (136.2)-7705-14-8	8.1	x			-	-				0.4			0.07
Menthone (154.3)-*89-80-5*	9.3		x^*)^		-	-		1.3	1.0	1.2		3.5	1.17
*iso*-Menthone (154.3)-*1196-31-2, 491-07-6*	9.4		x^*)^		-	-						1.3	0.22
Unknown 7	9.4	?	?	?	?	?		0.4	0.4	0.5			0.22
Menthol (156.3)-*89-78-1*	9.6	x			-	-		0.4	0.4	1.5		3.7	1.0
Carvone (150.2)-*99-49-0*	10.1		x		+^*)^	-						2.4	0.4
Unknown 8	10.1	?	?	?	?	?		0.7	0.6	0.6			0.32
*cis*-Anethole (148.2)-*25679-28-1*	10.2			x	-	-		2.1	2.4	0.4	1.3	1.4	1.27
Methyl salicylate (152.1)-*119-36-8*	10.3	x			-	-			1.4				0.23
*trans*-Anethole (148.2)-*4180-23-8*	10.5			x^*)^	-	-		8.7	2.2	12	18	28	11.5
2-Menthene (138.3)-*5113-93-9, 2230-85-5*	10.6	x			-	-						0.4	0.07
1-Menthene (138.3)-5502-88-5	10.6	x			-	-				0.2			0.03
*iso*-Eugenol (164.2)-*97-54-1* or	11.0	x			-	-						0.8	0.13
Eugenol (164.2)-*97-53-0*		x			+^*)^	-							
Anisketone (164.2)-*122-84-9*	11.2	x			-	-		2.0	2.2	0.9	0.7	1.6	1.23
Methyl parabene (152.2)-*99-76-3*	11.8	x			-	-						0.6	0.1

x: Cramer classification; ?: classification not possible due to unknown substance; -: negative alert; +: positive alert; *) THF has a SML = 0.6 mg/kg; carvone: wrong assignment by Toxtree, is not considered genotoxic (food additive); eugenol: food additive—FDA: GRAS – EU (FCM no. 180): SML = 50 ppb, not considered genotoxic; menthone, *iso*-menthone and anethole: EU and FDA food additives/flavouring agents.

**Table 4 molecules-25-04998-t004:** Substances found in PET bottles originally used for sanitary products.

Substance	GC R_t_	Cramer Classification					Concentration in PET [mg/kg]							
(mol.weight [g/mol])-	[min]	Class			Alert		Sample No.							
*CAS-No.*		I	II	III	gtc	ngtc	17	18	19	20	21	22	23	Mean 17–23
Ethanol (78.4)-*64-17-5*	1.8	x			-	-			78	438				73.7
Formic acid (46.0)-*64-18-6*	1.9	x			-	-						1.8	2.2	0.57
2-Butanone (79.6)-78-93-3	2.1	x			-	-			4.6	18			120	20.7
Tetrahydrofuran (72.1)-*109-99-9*	2.3		x^*)^		-	-					4.9	2.5		1.06
2-Methyl-1,3-dioxolane (88.1)-497-26-7	2.4	x			-	-	1.3	1.3	0.8	1.2	0.6	1.3		0.93
*n*-Butanol (74.1)-71-36-3	2.5	x			-	-		0.3						0.04
2-Ethylacrolein (84.1)-*922-63-4*	2.5		x		+^*)^	-					0.4	0.6		0.14
Ethylene glycol (62.1)-*107-21-1*	2.7	x			-	-	4.4	1.5	3.4	5.2	5.0	4.2		3.39
Hexanal (100.2)-*66-25-1*	4.3	x			+^*)^	-							2.9	0.41
Heptanal (114.2)-*111-71-7*	6.1	x			+^*)^	-							1.0	0.14
1-Butoxy-2-propanol (132.2)-*5131-66-8*	6.8	x			-	-				32				4.57
Unknown 9	6.9	?	?	?	?	?						0.9		0.13
Unknown 10	7.1	?	?	?	?	?				0.7				0.10
Octanal (128.2)-*124-13-0*	7.6	x			+^*)^	-							1.9	0.27
Unknown 11	7.6	?	?	?	?	?						1.8		0.26
Limonene (136.2)-*7705-14-8*	8.1	x			-	-			3.4	7.4		7.9		2.67
Unknown 12	8.4	?	?	?	?	?						3.0		0.43
Unknown 13	8.5	?	?	?	?	?							0.9	0.13
Nonanal (128.2)-124-13-0	8.8	x			+^*)^	-							2.9	0.41
1-Butoxypropan-2-ol (132.2)-*5131-66-8*	8.9	x			-	-				2.4				0.33
Unknown 14	9.1	?	?	?	?	?						1.6		0.22
Acetic acid benzylester (150.2)-*140-11-4*	9.3	x			-	-			3.2					0.46
Menthone (154.3)-*89-80-5*	9.3		x^*)^		-	-	1.8							0.26
*iso*-Menthone (154.3)-*1196-31-2, 491-07-6*	9.4		x^*)^		-	-	0.6							0.09
Menthol (156.3)-*89-78-1*	9.6	x					2.9							0.41
Unknown 15	9.1	?	?	?	?	?							4	0.55
4-Methoxy benzaldehyde (136.2)-*123-11-5*	10.1	x			x^*)^	-	1.9							0.27
*cis*-Anethole (148.2)-*25679-28-1*	10.2			x	-	-	31							4.43
*trans*-Anethole (148.2)-*4180-23-8*	10.5			x^*)^	-	-	44							6.28
Anisketone (164.2)-*122-84-9*	11.2	x			-	-	59							8.4

x: Cramer classification; ?: classification not possible due to unknown substance; -: negative alert; +: positive alert; *) THF has a SML = 0.6 mg/kg; 2-ethylacrolein is mutagenic in vitro (AMES Test) [19]; hexanal, heptanal, octanal and nonanal: wrong assignments by Toxtree, not genotoxic (flavouring agent); menthone, *iso*-menthone, 4-methoxybenzaldehyde (anisaldehyde), anethole: EU and FDA food additives/flavouring agents.

**Table 5 molecules-25-04998-t005:** Substances found in PET bottles originally used for shampoo products.

Substance	GC R_t_	Cramer Classification					Concentration in PET [mg/kg]						
(mol.weight [g/mol])-	[min]	Class			Alert		Sample No.						
*CAS-No.*		I	II	III	gtc	ngtc	24	25	26	27	28	29	Mean 24–29
Ethanol (78.4)-*64-17-5*	1.8	x			-	-						1110	185
2-Butanone (79.6)-*78-93-3*	2.1	x			-	-		4.2					0.70
2-Methyl-1,3-dioxolane (88.1)-*497-26-7*	2.4	x			-	-	1.4		2.5	2.3	1.7	0.7	1.44
Ethylene glycol (62.1)-*107-21-1*	2.7	x			-	-	3.6		3.2	4.9	4.6	4.5	3.47
Benzylalcohol (108.1)-*100-51-6*	7.9	x			-	-					8.9		1.48
Unknown 16	7.9	?	?	?	?	?						1.5	0.25
Limonene (136.2)-*7705-14-8*	8.1	x			-	-						2.9	0.48
Acetic acid, benzylester (150.2)-*140-11-4*	9.3	x			-	-		2.2	0.9				0.52
2-Hydroxybiphenyl (170.2)-*90-43-7*	12.2	x			-	+^*)^				1.8			0.30
Unknown 17	12.6	?	?	?	?	?						0.7	0.12

x: Cramer classification; ?: classification not possible due to unknown substance; -: negative alert; +: positive alert; *) 2-Hydroxybiphenyl (o-phenyl phenol): FDA substance added to food (surface active agent; WHO/JECFA: ADI up to 0.4 mg/kg b.w.

**Table 6 molecules-25-04998-t006:** Substances found in PET bottles originally used for shower gel or liquid soap products.

Substance	GC R_t_	Cramer Classification					Concentration in PET [mg/kg]							
(mol.weight [g/mol])-	[min]	Class			Alert		Sample No.							
*CAS-No.*		I	II	III	gtc	ngtc	30	31	32	33	34	35	36	Mean 30–36
Ethanol (78.4)-*64-17-5*	1.8	x			-	-	270	400	100					110
Unknown 18	1.8	?	?	?	?	?			14					2.0
2-Butanone (79.6)-*78-93-3*	2.1	x			-	-			13					1.86
2-Methyl-1,3-dioxolane (88.1)-497-26-7	2.4	x			-	-		0.8		2.2	1.0	0.7	1.5	0.89
Ethylene glycol (62.1)-*107-21-1*	2.7	x			-	-	3.1	4.7		4.3	5.9	5.0	2.7	3.67
Toluene (92.1)-*108-88-3*	3.9			x	-	-		3.4						0.49
Unknown 19	4.3	?	?	?	?	?			2.3					0.32
2,2-dimethyl-1,3-dioxolane (102.1)-*2916-31-6*	5.0			x	-	-	7.8							1.11
Benzaldehyde (106.1)-100-52-7	5.8	x			+^*)^	-				0.9				1.29
Unknown 20	6.1	?	?	?	?	?			4.4					0.63
Unknown 21	6.3	?	?	?	?	?	2.1							0.30
Benzylalcohol (108.1)-*100-51-6*	7.9	x			-	-				13				1.86
Limonene (136.2)-*7705-14-8*	8.1	x			-	-			9.5	2.0				1.64
Unknown 22	8.9	?	?	?	?	?							1.6	0.22
Unknown 23	9.3	?	?	?	?	?							0.9	0.13
Benzoic acid (122.1)-*65-85-0*	9.3	x			-	-	2.9				3.5	3.2		0.96
2-(Benzyloxy)ethanol (152.2)-*622-08-2*	9.9		x		-	-					2.0			0.29
Hexadecene (224.4)-*629-73-2*	12.9	x			--	-		1.4						0.20

x: Cramer classification; ?: classification not possible due to unknown substance; -: negative alert; +: positive alert; *) wrong assignment by Toxtree: benzaldehyde is not a human carcinogen and is not genotoxic.

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
