# Peer review of "Contamination Levels in Recollected PET Bottles from Non-Food Applications and their Impact on the Safety of Recycled PET for Food Contact"

_molecules, 2020, doi:10.3390/molecules25214998_

Round 1
Reviewer 1 Report
I have reviewed the manuscript MOLECULES-945209 entitled “Contamination levels in re-collected PET bottles from non-food applications and their impact on the safety of recycled PET for food contact”. I recommend that the paper should be accepted following the suggestions:
Reviewer decision: MAJOR REVISION
The paper offers an interesting study that improves the knowledge of concentration of post-consumer substances in the washed PET flakes originating from PET containers which had been used for non-food fillings.
An exhaustive bibliographic revision must be carried out in order to include the most current references relative to the last 2 years since only one reference from the year 2020 appears related to the migration of substances in PET bottles.
The section “Materials and Methods” should be described before the results and discussion becoming section 2 instead of section 3. In addition, the information reported in lines 156-158 in which is described how the PET containers were grouped must be removed since it is also described in the “Materials and Methods” section (lines 541-544). All the manuscript must be revised to avoid repeating information in different sections.
The experimental procedure carried out for the extraction and quantification of compounds is not well described (lines 545-556). The experimental work must be described in detail to improve understanding by the reader. For example, in lines 552-553 “The concentrations of the n-alkanes were determined quantitatively by extraction with dichloromethane followed by gas chromatographic detection”. How was this extraction process carried out (time, temperature ...)? Describe it in more detail.
Author Response
Reviewer 1
I have reviewed the manuscript MOLECULES-945209 entitled “Contamination levels in re-collected PET bottles from non-food applications and their impact on the safety of recycled PET for food contact”. I recommend that the paper should be accepted following the suggestions:
Reviewer decision: MAJOR REVISION
The paper offers an interesting study that improves the knowledge of concentration of post-consumer substances in the washed PET flakes originating from PET containers which had been used for non-food fillings.
An exhaustive bibliographic revision must be carried out in order to include the most current references relative to the last 2 years since only one reference from the year 2020 appears related to the migration of substances in PET bottles.
Answer: We focussed our literature research on contaminants in PET bottles or washed PET flakes as input materials for recycling processes, especially on non-food bottles. The focus of the literature research was not on the migration from PET bottles in general. Most of the publications are published around the European Recyclability project which runs from 2001 to 2003. However, we made a new literature search and included two new citations.
The section “Materials and Methods” should be described before the results and discussion becoming section 2 instead of section 3.
Answer: As far as we understood, the Materials and Methods Section is behind the Discussion. Therefore, we made no changes made in the manuscript according to this point.
In addition, the information reported in lines 156-158 in which is described how the PET containers were grouped must be removed since it is also described in the “Materials and Methods” section (lines 541-544). All the manuscript must be revised to avoid repeating information in different sections.
Answer: Since the Materials and Methods section is behind the discussion, we think that it is necessary to mention the sample groups in both sections. No changes made in the manuscript.
The experimental procedure carried out for the extraction and quantification of compounds is not well described (lines 545-556). The experimental work must be described in detail to improve understanding by the reader. For example, in lines 552-553 “The concentrations of the n-alkanes were determined quantitatively by extraction with dichloromethane followed by gas chromatographic detection”. How was this extraction process carried out (time, temperature ...)? Describe it in more detail.
Answer: We include the experimental details in the materials and methods section.
Reviewer 2 Report
The work by Franz and Welle covers an important topic for the food industry, especially in light of the implications for the consumers' health. The work is original and with a high degree of novelty, which clearly shows the highly specialized background of the authors and their deep knowledge of the subject matter. The experimental part, and in particular the design of the experiments and the mathematical approach, provides an additional merit to this excellent work. Due the overall high quality, I recommend to accept the manuscript in the present form, provided that the authors strive to address the following minor points:
- simplify some parts of the manuscript; some sentences, while being clear, are too long (see for example lines 14-19), which make the overall reading a bit arduous;
- double-check some parts of the text for typos and missing words (see for example line 604: "Thiscriterion, however, appears to BE very conservative..."
- in the Introduction section, the author state that "...in average 58.2% of the marketed PET bottles were re-collected and recycled in 2017"; please specify if this information concerns food/beverage-contact applications or if it is an overall (irrespective of the final use) estimation;
- it would be nice for the readership to know whether the use of recycled PET for new food packaging materials is indeed widespread throughout Europe; in other words, is there any difference between EU countries? Is there any national legislation that somehow hinders (or regulate) this aspect in a specific way for each country? This is because, based on the reviewer's perception, in the southern-Europe countries the use of r-PET bottles is hardly seen; can you confirm this?
- in Figure 1, please replace commas with full stops in the figures of y-axis.
Author Response
Reviewer 2
The work by Franz and Welle covers an important topic for the food industry, especially in light of the implications for the consumers' health. The work is original and with a high degree of novelty, which clearly shows the highly specialized background of the authors and their deep knowledge of the subject matter. The experimental part, and in particular the design of the experiments and the mathematical approach, provides an additional merit to this excellent work. Due the overall high quality, I recommend to accept the manuscript in the present form, provided that the authors strive to address the following minor points:
simplify some parts of the manuscript; some sentences, while being clear, are too long (see for example lines 14-19), which make the overall reading a bit arduous;
Answer: Some sentences are indeed too long. We revised the manuscript accordingly.
double-check some parts of the text for typos and missing words (see for example line 604: "Thiscriterion, however, appears to BE very conservative..."
Answer: We revised the manuscript accordingly.
in the Introduction section, the author state that "...in average 58.2% of the marketed PET bottles were re-collected and recycled in 2017"; please specify if this information concerns food/beverage-contact applications or if it is an overall (irrespective of the final use) estimation;
Answer: 58.2% is the recollection rate in Europe for beverage bottles. We included "beverage bottles" in the manuscript.
it would be nice for the readership to know whether the use of recycled PET for new food packaging materials is indeed widespread throughout Europe; in other words, is there any difference between EU countries? Is there any national legislation that somehow hinders (or regulate) this aspect in a specific way for each country? This is because, based on the reviewer's perception, in the southern-Europe countries the use of r-PET bottles is hardly seen; can you confirm this?
Answer: This is indeed an intersting point. However, we collected our sample material in Germany. Samples from other countries are not collected and not analysed. Therefore, we cannot discuss this point because experimental data are not available. Regarding legislation, the recycling regulation 282/2008 in in force in all EU countries. Less recyclate amounts in southern countries is in our point of view due to less recollection rates or poor recyclate quality.
in Figure 1, please replace commas with full stops in the figures of y-axis.
Answer: This is an excel chart. We cannot change the commas into full stops.
Reviewer 3 Report
Presented manuscript aimed at determination of contaminants in re-collected PET bottles from non-food applications and their impact on the safety of recycled PET for food contact. Although the aim of research is interesting, manuscript is not proper organized. Many sections require expansion and a re-phrase. The use of English is not adequate. Some fragments were written rather using a popular language. Analytical aspects were not proper performed and described. Moreover, the Authors did not refer to the latest literature in this field. In most cases, the publications cited were published more than 10-20 years ago. Therefore, I do not recommend the manuscript for publication in the Molecules.
Although I do not recommend presented manuscript for publication, prepared some suggestions and advices for the Authors. All are listed below.
Abstract: The text is too long. According to the journal requirement, abstract should be no longer than 200 words.
- Introduction:
L31: Please change Poly (ethylene terephthalate) PET to Poly (ethylene terephthalate) (PET)
L58: remove the citation [4]. The next sentence L60 is also supported by ref [4] so only one ref needs to be using for both sentences.
L64: change [5, 6, 7, 8] to [5-8]. L64, L146 in the same way.
Introduction part is long but little informative. There is little information on other studies conducted and trends in this area of research. A review of current literature data should be added. Moreover, the novelty of this research should be highlighted at the end of the introduction part.
- Results and Discussion
Tab 1-6: Results should be rounded to 2 significant digits. Moreover, the results should be noticed as AV±SD.
Why were the results not given for all the tested samples? The table 1, for ethanol, only the result is given for one sample, and there were six tested samples. If there is no result in the table, it means that the analysis was not performed or the result was <LOD ? It should be clarified.
Why did the Authors use the mean value for all tested samples from the given category? On the ethanol example: If the concentration were determined in only one of the six tested samples (380 mg/kg), it cannot be estimated that for the remaining samples these concentration values are equal (63.3 mg/kg per every sample).
3.Materials and Methods
There is no information on the standards and reagents used during the studies. There is no information on the sample preparation procedure. Needs to be added.
3.2. Quantitative and semi-quantitative determination of chemicals in the PET containers
Please add more detailed information on methodologies used during this study. There are many references instead of description of procedures. It is very difficult to follow the whole procedure in the present form. Especially, the publications referenced have been published in 2004, 1982 and 1991, which raises great doubts as to the correctness of the applied methodology. Maybe, including these descriptions in supplementary materials would be helpful. Additionally, basic information on the standards preparation procedure and a validation parameter of the chromatographic procedure will be valuable. There is no information on the number of repetitions. Was the analysis carried out only in one repetition? ? This information must be added. Why was none of the obtained chromatograms shown in the paper?
3.3 Migration modelling
As above, please add some basic information instead ref.
Author Response
Reviewer 3
Presented manuscript aimed at determination of contaminants in re-collected PET bottles from non-food applications and their impact on the safety of recycled PET for food contact. Although the aim of research is interesting, manuscript is not proper organized. Many sections require expansion and a re-phrase. The use of English is not adequate. Some fragments were written rather using a popular language. Analytical aspects were not proper performed and described. Moreover, the Authors did not refer to the latest literature in this field. In most cases, the publications cited were published more than 10-20 years ago. Therefore, I do not recommend the manuscript for publication in the Molecules.
Answer: We thoroughly revised the manuscript and a native speaker made an English check. As mentioned above focussed our literature research on contaminants in PET bottles or washed PET flakes as input materials for recycling processes, especially on non-food bottles. We made again a literature search and to our knowledge we cited the relevant literature.
Although I do not recommend presented manuscript for publication, prepared some suggestions and advices for the Authors. All are listed below.
Abstract: The text is too long. According to the journal requirement, abstract should be no longer than 200 words.
Answer: The abstract in the submitted manuscript has 219 words. According to the guidance for authors, the abstract should have "about 200 words". So we are in this range. Therefore, we made no changes in the manuscript.
Introduction:
L31: Please change Poly (ethylene terephthalate) PET to Poly (ethylene terephthalate) (PET)
Answer: We change to (PET)
L58: remove the citation [4]. The next sentence L60 is also supported by ref [4] so only one ref needs to be using for both sentences.
Answer: We removed the first [4] citation
L64: change [5, 6, 7, 8] to [5-8]. L64, L146 in the same way.
Answer: This point is in the responsibility of the journals final check. Citations are introduced as cross links, which does not allow [5-8], because 6 and 7 will be eliminated.
Introduction part is long but little informative. There is little information on other studies conducted and trends in this area of research. A review of current literature data should be added. Moreover, the novelty of this research should be highlighted at the end of the introduction part.
Answer: As mentioned above, we focussed our literature search on contaminants in non-food bottles or substances from the first use found in washed PET recyclates (flakes). The literature here is indeed old, published in the 2000ff years when PET bottle to bottle recycling had been established.
Results and Discussion
Tab 1-6: Results should be rounded to 2 significant digits. Moreover, the results should be noticed as AV±SD.
Answer: We analysed only on bottle in triplicate. In our point of view AV±SD makes no sense, when only one bottle had been analysed. The standard deviation from the measurements in triplicate will be the analytical tolerance of the equipment and not of the samples. Therefore, we made no changes in the manuscript.
Why were the results not given for all the tested samples? The table 1, for ethanol, only the result is given for one sample, and there were six tested samples. If there is no result in the table, it means that the analysis was not performed or the result was <LOD ? It should be clarified.
Answer: In order to make the tables clearer, we do not include <LOD. In case of empty lines, the substance was not detected in the sample. We included a paragraph in the experimental section of the manuscript.
Why did the Authors use the mean value for all tested samples from the given category? On the ethanol example: If the concentration were determined in only one of the six tested samples (380 mg/kg), it cannot be estimated that for the remaining samples these concentration values are equal (63.3 mg/kg per every sample).
Answer: As mentioned in the manuscript (lie 166-169 in the revised manuscript), "mean values would form the right basis for further exposure and safety assessment considerations for real-life scenarios. However, for numerous substances this would lead to "dilutions" of concentrations in the full mixture. To avoid this, our approach is to undertake a safety evaluation based on typical concentrations and related ranges found in the six groups." If some substances are detected only in one sample, this will be also the case in real recollection systems. After recycling, the flake samples are homogeneous in their concentrations. Therefore, the mean value is in our point of view the right value for exposure evaluation. No changes made in the manuscript.
3.Materials and Methods
There is no information on the standards and reagents used during the studies. There is no information on the sample preparation procedure. Needs to be added.
Answer: We included the methods into the manuscript.
3.2. Quantitative and semi-quantitative determination of chemicals in the PET containers
Please add more detailed information on methodologies used during this study. There are many references instead of description of procedures. It is very difficult to follow the whole procedure in the present form. Especially, the publications referenced have been published in 2004, 1982 and 1991, which raises great doubts as to the correctness of the applied methodology. Maybe, including these descriptions in supplementary materials would be helpful. Additionally, basic information on the standards preparation procedure and a validation parameter of the chromatographic procedure will be valuable. There is no information on the number of repetitions. Was the analysis carried out only in one repetition? ? This information must be added. Why was none of the obtained chromatograms shown in the paper?
Answer: We included the methods into the manuscript.
3.3 Migration modelling
As above, please add some basic information instead ref.
Answer: In the chapter migration modelling only the applied software is described. Basic information on migration modelling would go beyond the scope of this study.
Round 2
Reviewer 1 Report
In my opinion, the article can be published in Molecules in the present form
Best regards
Ana
Author Response
Thanks
Reviewer 3 Report
The manuscript has been greatly improved; however, method is wrongly assumed, hence no validation data is presented.
Whole set of data cannot be correlated with any existing one due to the fact that Authors failed to present reliable data covering method development and sample preparation.
There is no proof that would emphasize the trueness of the obtained concentration of analytes in samples of PET bottles.
Author Response
The intention of the paper is not to provide a new method for the determination of contaminants in PET recyclates. We used the same approach which was previously used to screen washed PET recyclates within the European "Recyclability" Project. The method has not been changed to ensure the comparability of results.
We included "... within the European "Recyclability" Project [13]. The method has not been changed to ensure the comparability of results." i n the experimental section in order to make this clear.